# Plasmids Expressing shRNAs Specific to the Nucleocapsid Gene Inhibit the Replication of Porcine Deltacoronavirus In Vivo

**DOI:** 10.3390/ani11051216

**Published:** 2021-04-23

**Authors:** Jun Gu, Hao Li, Zhen Bi, Kai Li, Zhiquan Li, Deping Song, Zhen Ding, Houjun He, Qiong Wu, Dongyan Huang, Ping Gan, Yu Ye, Yuxin Tang

**Affiliations:** 1Department of Preventive Veterinary Medicine, College of Animal Science and Technology, Jiangxi Agricultural University, Nanchang 330045, China; gujun2021@outlook.com (J.G.); lllllllhao@sina.cn (H.L.); biz0110@126.com (Z.B.); lk15907007548@126.com (K.L.); lizhiquan2017@outlook.com (Z.L.); sdp8701@jxau.edu.cn (D.S.); dingzhenhuz@163.com (Z.D.); hehoujun@163.com (H.H.); lbls2005@sina.com (Q.W.); huangdongyan@jxau.edu.cn (D.H.); tang53ster@gmail.com (Y.T.); 2Jiangxi Engineering Research Center for Animal Health Products, Nanchang 330045, China; 3Guilin Liyuan Grain and Oil Food Group Co., Ltd., Guilin 541012, China; 4Animal Diseases Control and Prevention Center of Jiangxi Province, Nanchang 330006, China; jxcadcgp@163.com

**Keywords:** porcine deltacoronavirus, RNA interference, short hairpin RNA, nucleocapsid

## Abstract

**Simple Summary:**

Porcine deltacoronavirus (PDCoV) is an emerging enteropathogen distributed globally, which causes substantial economic losses in the swine industry. The characterization of the receptor promiscuity may pose a risk of cross-species transmission. However, the options for pharmaceutical interventions are limited. In this study, the vectors expressing shRNAs targeting the nucleocapsid gene were generated to assess the inhibition effect of PDCoV reproduction. Our preliminary results demonstrate that a dual shRNA expression system is an effective strategy in combating PDCoV infection without cytotoxicity, which would facilitate the ongoing development of RNAi-based therapeutic drugs against viral diseases.

**Abstract:**

Porcine deltacoronavirus (PDCoV) is a novel enteric coronavirus and is becoming one of the major causative agents of diarrhea in pig herds in recent years. To date, there are no commercial vaccines or antiviral pharmaceutical agents available to control PDCoV infection. Therefore, developing a reliable strategy against PDCoV is urgently needed. In this study, to observe the antiviral activity of RNA interference (RNAi), four short hairpin RNAs (shRNAs) specific to the nucleocapsid (N) gene of PDCoV were designed and tested in vitro. Of these, a double-shRNA-expression vector, designated as pSil-double-shRNA-N1, was the most effectively expressed, and the inhibition of PDCoV replication was then further evaluated in neonatal piglets. Our preliminary results reveal that plasmid-based double-shRNA-expression targeting the N gene of PDCoV can significantly protect LLC-PK1 cells and piglets from pathological lesions induced by PDCoV. Our study could benefit the investigation of the specific functions of viral genes related to PDCoV infection and offer a possible methodology of RNAi-based therapeutics for PDCoV infection.

## 1. Introduction

Based on antigenic relationships [1], coronaviruses, members of the *Coronaviridae* family, are conventionally composed of three genera, that is, *Alphacoronavirus*, *Betacoronavirus*, and *Gammacoronavirus*. Recently, a novel genus, *Deltacoronavirus*, has been discovered in varied host species, including birds and mammals [2,3]. In a survey conducted in Hong Kong, deltacoronaviruses were first identified in both avian and porcine specimens in 2012 [1]. Thereafter, a porcine deltacoronavirus (PDCoV) associated with diarrhea was first reported in Ohio in 2014, and then detected in many other states of the United States [4,5,6,7]. PDCoV has since been prevalent in swine herds in pig-raising countries all over the world, leading to substantial economic losses [8,9,10,11,12,13]. Previous reports have indicated that host aminopeptidase N (APN) is an entry receptor for PDCoV infection, which makes some mammalian and avian cells susceptible to PDCoV. In fact, it has been proven that PDCoV is able to infect chickens and calves. Broad receptor engagement of PDCoV makes it potentially able to breach the species barrier between birds and mammals [14], which might have public health significance.

PDCoV is an enveloped, positive-sense, single-stranded RNA virus. The genome of PDCoV consists of 5′ untranslated region (UTR), seven open reading frames (ORFs), and 3′ UTR, encoding at least four nonstructural proteins (replicase polyproteins 1a (pp1a), pp1ab, nonstructural protein 6 (NS6), and nonstructural protein 7 (NS7)), and four structural proteins (spike (S), envelope (E), membrane (M), and nucleocapsid (N) proteins) [1]. The biological properties of the structural and nonstructural proteins of coronaviruses in the genera other than deltacoronavirus have been well-documented [15,16]. Of the structural proteins of coronaviruses, the N protein abundant in infected cells is a multifunctional protein that forms viral core complexes with genomic RNA, is essential for the virus life cycle, and plays an important role in enhancing the efficiency of virus transcription and assembly [17]. PDCoV is a newly emerged coronavirus, and thus, little is known in terms of the biological properties of the N protein in viral replication and host immune modulation. A recent study reported that the N protein was distributed in both the nucleus and cytoplasm, indicating that the N protein may be involved in RNA synthesis and/or ribosome biogenesis through its interactions with nuclear proteins and/or ribosomal subunits of host cells [18,19,20]. Notably, the N protein causes significant alterations in the expression of cellular proteins associated with the metabolic process, cell division, protein synthesis and transportation, cytoskeleton networks and cell communication, and stress response. Currently, a commercial PDCoV vaccine is not available. Thus, an antiviral alternative is valuable in terms of the prevention and control of PDCoV infection.

RNA interference (RNAi) is a sequence-specific gene suppression mechanism triggered by short complementary double-stranded RNA molecules. Short hairpin RNA (shRNA)-based technology has become a benchmark tool to induce RNAi, which is characterized by base-paired stems and a loop region [17,21]. In recent years, RNAi technology has been successfully developed as a new antiviral therapeutic strategy, and widely applied to inhibit viral replication in vitro and in vivo, including against human immunodeficiency virus type 1 (HIV-1) [22], severe acute respiratory syndrome coronavirus (SARS-CoV) [23], influenza virus [24], classical swine fever virus (CSFV) [25], foot-and-mouth disease virus (FMDV) [26], transmissible gastroenteritis virus (TGEV) [27], and porcine epidemic diarrhea virus (PEDV) [28].

In this study, four shRNAs targeting the N gene of PDCoV were designed and screened to evaluate the protective effects of RNAi against PDCoV infection. The recombined plasmid of a double-shRNA-expression, designated as pSil-double-shRNA-N1, was the most effective shRNA in the suppression activity on PDCoV proliferation in LLC-PK1 cells and piglets. The results suggest that pSil-double-shRNA-N1 inhibited PDCoV replication significantly without cytotoxicity in vitro and in vivo. Our findings in this preliminary study increase the knowledge of the ongoing development of RNAi-based antivirals, and thus provide an alternative for the prevention and control of PDCoV infection.

## 2. Materials and Methods

### 2.1. Ethical Statement

The ethics committee of Jiangxi Agricultural University approved the animal use protocol for this study (Number: JXAU-AE-2018-25). All procedures involving animals in this study were conducted according to the Guidelines for the Care and Use of Experimental Animals established by the Ministry of Agriculture and Rural Affairs of China.

### 2.2. Virus and Cells

PDCoV strain CH/JXNI/02/2015 was isolated from a diarrheal piglet and stored in our laboratory [12]. A LLC-PK1 cell line was obtained from the Cell Resource Center, Peking Union Medical College, the headquarters of the National Infrastructure of Cell Line Resource (NSTI), and grown in high glucose Dulbecco Minimum Essential Medium (DMEM) with the addition of 10% fetal bovine serum (Gibco, Gaithers-burg, MD, USA), 100 IU/mL of penicillin, and 100 μg/mL of streptomycin, in a 37 °C incubator with 5% CO_2_.

### 2.3. shRNA Design and Selection

The N gene sequence of CH/JXNI/02/2015 was analyzed using the online siRNA target design tool to choose the candidate target sequences (https://www.thermofisher.com/cn/zh/home/life-science/rnai/synthetic-rnai-analysis/stealth-rnai-technology.html (accessed on 10 April 2018). There were 4 potential target shRNAs designed, specific to different conserved nucleotide positions of the N gene of PDCoV (Table 1). The specificity of the sequences was then examined by BLAST to ensure that they would not hit any sequences of the swine genome, and had 100% identity with the targeted gene.

### 2.4. Construction of shRNA Plasmids

The protocol for the construction of shRNA plasmids was described previously [27,28]. Briefly, sense and antisense DNA oligonucleotides dissolved in sterile ddH_2_O were annealed in 25 μL of reaction system containing 5 μL (100 μM) of sense (forward) oligonucleotide, 5 μL (100 μM) of antisense (reverse) oligonucleotide, and 15 μL of ddH_2_O. The mixture was heated to 95 °C for 5 min, then cooled down to 50 °C and kept for 30 s, and subsequently incubated at 4 °C for 30 min. There were 2 shRNA-expressing plasmids constructed, pSil-U6-mcherry and pSil-double-U6-mcherry, as mentioned before [29], and then transformed into *Escherichia coli* DH5a competent cells. The positive clones were selected and then validated by PCR amplification and DNA sequencing.

### 2.5. Generation of LLC-PK1 Cells Stably Expressing shRNA and Virus Infection

The LLC-PK1 cells were seeded (2 × 10^4^/well) into 6-well plates and incubated for 24 h at 37 °C in a 5% CO_2_ atmosphere. When reaching 50–70% confluence, the cells were washed three times with sterilized 0.01 M pH7.4 phosphate buffered saline (PBS). Afterwards, the cells were transfected with 2.5 μg/well of shRNA-expressing plasmids using a Lipofectamine^TM^ 3000 (Invitrogen, Waltham, MA, USA), according to the manufacturer’s instructions. After 24 h of incubation, the growth media were substituted with maintenance media containing 2% FBS and 1000 μg/mL of Neomycin (G418). The survival cell clones were maintained in G418-containing media for 15 d with frequent media replacements until cell death could no longer be observed. Then, these monoclonal cells transfected with shRNA-expressing plasmids were screened by limiting dilution analysis (LDA), as previously described [29], and cultured in DMEM growth media containing G418 (500 μg/mL) in 6-well plates at 37 °C in a 5% CO_2_ atmosphere. After reaching a confluence, these plasmid-transduced cells were inoculated with PDCoV at a multiplicity of infection (MOI) of 0.1. Non-transfected cells were set as a control. Cell transfection efficiency and cytopathic effect (CPE) images were taken under an inverted fluorescence/phase-contrast microscope (Nikon, Tokyo, Japan).

### 2.6. Cell Viability Determination (MTS Assay)

The constructed stable cell lines of LLC-PK1 with shRNA expression were seeded into 96-well plates at a density of 1 × 10^4^/well, and then treated as described above. The cells were infected with PDCoV at an MOI of 0.1. Cell viability was determined by CellTiter 96^®^ AQueous One Solution Cell Proliferation Assay (Promega, Madison, WI, USA) according to the manufacturer’s protocol. Briefly, 20 µL/well of the MTS reagent [3-(4,5-dimethylthiazol-2-yl)-5-(3-carboxymethoxyphenyl)-2-(4-sulfop henyl)-2H-tetrazolium] was added at 36 h post-infection (hpi) and the cells were incubated at 37 °C for 4 h in a 5% CO_2_ atmosphere. The absorbance at 490 nm of each solution was read by a Varioskan Flash Spectral Scanning Multimode Reader (Thermo Fisher, Waltham, MA, USA). The cell viability was calculated as a percentage of the OD value of the treated cells versus the control cells. All experiments were conducted in triplicate.

### 2.7. Virus Titration (TCID_50_ Determination)

The stable cell lines of LLC-PK1 with shRNA expression were harvested at 48 hpi and subjected to freeze–thaw three times. After clarification based on the standard methodology, the cell culture supernatants were collected and made into a 10-fold serial dilution (10^−1^ to 10^−10^), and then added into 96-well plates pre-seeded with LLC-PK1 cells to determine the virus titers expressed as TCID_50_, which was determined using the Reed–Muench method [30].

### 2.8. RNA Extraction and Quantitative Real-Time PCR (qRT-PCR)

To estimate the impact of shRNA on viral replication, a qRT-PCR assay was used. For this purpose, total RNA was initially extracted from the LLC-PK1 cells at 36 hpi using the MiniBEST™ Viral RNA/DNA Extraction Kit Ver.5.0 (Takara, Kyoto, Japan), and then reversely transcribed into the first-strand cDNA using the GoScript^TM^ Reverse Transcription System (Promega, Madison, WI, USA), following the manufacturer’s instructions. The qRT-PCR of PDCoV genomic RNA was performed in a total volume of 20 μL reaction mixture comprising 10 μL of SYBR Premix Ex Taq II (Takara, Kyoto, Japan), 0.4 μL of forward primer, 0.4 μL of reverse primer, 2 μL of cDNA template, and 0.4 μL of Rox Reference Dye, and the β-actin gene was employed as an internal reference control (Table 2). The qRT-PCR was performed on an ABI 7500 Real-Time PCR System (Thermo Fisher, Carlsbad, CA, USA) under the following parameters: initial denaturation at 95 °C for 30 s, and then 40 cycles of 95 °C for 5 s and 61 °C for 30 s; the melting curve stage comprised 95 °C for 1 min, 55 °C for 30 s, and 95 °C for 30 s. Each experiment was repeated three times. The relative expression level of the N gene of PDCoV in infected cells was estimated by the 2^−△△Ct^ method [28].

### 2.9. Western Blotting

The stable LLC-PK1 cell lines expressing shRNA were infected with PDCoV at an MOI of 0.1. The infected cells were collected at 36 hpi and washed three times with cold PBS (0.01 M, pH7.4), following the aforementioned procedures. The cell pellets were lysed in radio immunoprecipitation assay (RIPA) lysis buffer with a protease inhibitor phenylmethane sulfonyl fluoride (PMSF) (Beyotime, Shanghai, China). Each sample was denatured in 5 × loading buffer at 100 °C for 10 min. The total proteins were separated using sodium dodecyl sulfate-polyacrylamide gel electrophoresis (SDS-PAGE) and electrotransferred onto a polyvinylidene difluoride (PVDF) membrane (Millipore, Billerica, MA, USA). The membranes were incubated with rabbit anti-PDCoV polyclonal antibodies prepared in our laboratory at 4 °C overnight, washed three times with 0.01 M pH 7.4 PBS containing 0.05% Tween-20 (PBST), and then incubated with HRP-conjugated goat anti-rabbit antibody (Transgen, Beijing, China) for 2 h at room temperature. The membranes were washed with PBST and visualized by enhanced chemiluminescence reagents and recorded on a Chemi-Doc imaging system (Bio-Rad, Hercules, CA, USA).

### 2.10. Animal Challenge Experiments

There were 12 healthy colostrum-deprived (CD) piglets randomly divided into 6 groups (2 piglets for each group; Table 3). Each piglet was injected via the pre-cava with 3 mL of pSil-double-shRNA-N1-mcherry or pSil-shRNA-NC-mcherry at a dose of 3 mg/pig and an equal volume of phosphate buffered saline (PBS), respectively. At 48 h post-injection, all piglets in groups P + V, N + V, and C + V were inoculated with 10^5^ TCID_50_ PDCoV orally, while groups P, N, and C were non-infection controls. The clinical signs of the piglets were monitored every 4 h before 12 h of PDCoV infection, including appetite, body weight, diarrhea, and vomiting. Fecal samples were collected by rectal swabs for qRT-PCR analysis at the same time points. All piglets were then euthanized and necropsied to examine the macroscopic lesions, and the small intestines were harvested at 48 hpi. The paraffin-embedded duodenum, jejunum, and ileum tissues were processed for histopathological analysis with hematoxylin and eosin (H&E) staining. The length of the villi and crypts of each small intestine segment were measured respectively and the villous height-to-crypt depth (V/C) ratios were further calculated.

### 2.11. Statistical Analysis

The data were statistically analyzed in GraphPad Prism Software version 5.01 (GraphPad Software, San Diego, CA, USA). The data are presented as the mean ± SD for all experiments. A two-tailed t-test was used to assess the significance of the differences between the means. A *p*-value of <0.05 (*) was considered significantly different and a *p*-value of <0.001 (***) was considered highly significant.

## 3. Results

### 3.1. Generation of LLC-PK1 Cell Lines Stably Expressing shRNA and Exploration of the Antiviral Properties of shRNA

On account of the low transfection efficiency of plasmid DNA, LLC-PK1 cell lines stably expressing shRNA were established in this study. As illustrated in Figure 1, the high red fluorescent intensity was observed in cells containing shRNA recombinant plasmids pSil-shRNA-N1-mCherry, pSil-shRNA-N2-mCherry, pSil-shRNA-N3-mCherry, pSil-shRNA-N4-mCherry, and pSil-shRNA-NC-mCherry, while the normal LLC-PK1 cells showed no fluorescence. To screen the inhibition potency of shRNA against PDCoV, the LLC-PK1 cells stably expressing shRNA were infected with PDCoV at an MOI of 0.1. The cells in the infected positive control group and the pSil-shRNA-NC-mCherry-transfected group exhibited pronounced cytopathology at 36 hpi, including cell syncytium formation, shrinkage, and death when compared with the mock-infected cells that remained undetached to the bottom surfaces of the plate wells and maintained their normal shapes. In contrast, the cells harboring the shRNA-expressing plasmids pSil-shRNA-N1-mCherry, pSil-shRNA-N2-mCherry, and pSil-shRNA-N3-mCherry rarely showed CPE. However, the cells harboring pSil-shRNA-N4-mCherry displayed severe CPE. The results of the qRT-PCR and Western blotting also demonstrate that pSil-shRNA-N1-mCherry, pSil-shRNA-N2-mCherry, and pSil-shRNA-N3-mCherry significantly inhibited genomic RNA replication and N protein expression of PDCoV, while pSil-shRNA-N4-mCherry displayed weaker suppression capability (Figure 2 and Figure 3). Among them, shRNA-N1 possessed the most robust antiviral performance. Hence, shRNA-N1 was selected to establish tandem expression vector pSil-double-shRNA-N1-mcherry for further study.

### 3.2. Anti-PDCoV Activity of pSil-Double-shRNA-N1-Mcherry in Infected LLC-PK1 Cells

To examine if there was enhanced inhibition potential of double-shRNA-expressing plasmids during PDCoV infection, the LLC-PK1 cell lines expressing pSil-double-shRNA-N1-mcherry were developed. After inoculation with PDCoV for 36 h, the typical CPE was observed microscopically in infected control cells, whereas no apparent CPE was found in cells expressing double shRNA-N1 sequences (Figure 4). Meanwhile, PDCoV RNA amounts and N protein synthesis in the samples from pSil-double-shRNA-N1-mcherry group were dramatically reduced as compared with the samples from the single shRNA-N1 plasmid group (Figure 5 and Appendix A). The virus titers in the LLC-PK1 cell lines with double shRNAs co-expressing were significantly lower when compared with the infected control, which corresponded to approaching over a 2154-fold drop (Figure 6).

To further access the cell viability under the influence of shRNAs during PDCoV infection, an MTS assay was performed on the LLC-PK1 cell lines containing shRNA plasmids. As shown in Figure 7, pSil-double-shRNA-N1-mcherry was highly effective in protecting the LLC-PK1 cells against PDCoV compared with the infected control, with a cell viability (%) of 81.624 ± 0.024%. However, pSil-shRNA-NC-mCherry could not exert similar impact, only keeping 21.525 ± 0.010% cells alive. These data uncovered that pSil-double-shRNA-N1-mcherry exhibited strong potency against PDCoV infection. Remarkably, when compared with the mock control, the cell viability contributed by pSil-shRNA-NC-mCherry and pSil-double-shRNA-N1-mcherry was 96.68 ± 0.025% and 88.099 ± 0.031%, respectively, which was not significantly different from the mock control, indicating that the introduction of shRNA into LLC-PK1 cells did not induce obvious cytotoxicity.

### 3.3. Inhibition Effects of pSil-Double-shRNA-N1-Mcherry Plasmids on PDCoV Replication in Neonatal Piglets

To observe the anti-PDCoV effects of the double-shRNAs-expressing plasmids in vivo, the piglets were given paravertebral injections with pSil-double-shRNA-N1-mcherry plasmids, and then challenged with PDCoV. At 36 hpi, the challenged neonatal piglets in groups N + V and C + V started showing depression, anorexia, watery diarrhea, dehydration, and weight loss, and the characteristic gross lesions were discovered in the small intestines (duodenum, jejunum, and ilium), including transparent and thin intestinal walls, mesenteric congestion, and hemorrhage (Figure 8). In contrast, no clinical symptoms were observed in the piglets from the P + V and non-infection groups; the only pathological changes encountered in postmortem examination at 48 hpi was an intermediate level of yellow fluid accumulation in the small intestine lumen. Histopathological analysis demonstrated moderate villous atrophy in the PDCoV-infected piglets after pSil-double-shRNA-N1-mcherry administration, in which the villus-to-crypt (V/C) ratios of each intestine segment were significantly higher than those of the infection control (Table 4 and Appendix A). Notably, the average V/C ratios of the double-shRNA-N1 plasmid group and the shRNA-NC plasmid groups were relatively lower when compared with those of the mock control, indicating that shRNA-expressing plasmids might not completely protect the infection caused by PDCoV, or perhaps had mild cytotoxicity toward the piglets. However, the histological lesions in the P + V group were not more pronounced than those of the positive control group (Figure 9). Therefore, our results evidence that double expression shRNAs specific to the N gene could alleviate the intestinal injury caused by PDCoV infection. Most importantly, as displayed in Figure 10, fecal virus shedding in the infection control and the shRNA-NC + PDCoV groups presented similar patterns of high viral shedding, while the piglets treated with double-shRNA-N1 had markedly lower viral loads, indicating that pSil-double-shRNA-N1-mcherry could effectively inhibit PDCoV replication in vivo.

## 4. Discussion

Widely used for gene regulation in mammalian and human cells [31,32,33], RNAi is a conserved biological response to small dsRNA that initiates sequence-specific gene suppression/silencing at the post-transcriptional level [17]. RNAi can be delivered into the cells via two different ways, that is, either through exogenously synthetic siRNAs or plasmid/viral vectors encompassing shRNA−encoding fragments. Although synthesized siRNAs can effectively induce the degradation of targeted RNA in transfected cells, the effects are transient. The plasmids or virus vectors-based shRNAs are much more stable than RNAs in certain environments and can be processed by a host cellular mechanism to continuously produce siRNAs in the host cells, which can overcome the disadvantages of chemically synthesized siRNAs [21]. Specifically, viral vectors, including lentiviruses, adenovirus, and other self-replicating RNA viruses, could trigger long-term inhibition activity [34].

At present, the threat posed by viral infections remains a serious challenge due to the emergence of novel viral pathogens and reemergence of variant strains of known virus species. For fighting viral diseases with RNAi, considerable attempts have been made to date. Previous reports demonstrated that the plasmid vector constructs containing shRNAs targeting CSFV [25], FMDV [26], TGEV [27], and PEDV [28] could effectively inhibit viral replications [35,36,37,38]. RNAi-based technology is becoming a potent therapeutic strategy to exert antiviral activity [39].

PDCoV is a newly emerged enteric pathogen causing severe diarrheal disease and mortality in neonatal piglets, which has resulted in huge economic losses to the global pig industry. PDCoV has access to a variety of host cells, which implies that it may have a propensity for cross-species transmissibility [14]. To date, it remains a serious problem to efficiently prevent the spread of PDCoV infection. As known, the N protein of coronaviruses is a major structural protein that is involved in virus assembly, and plays a critical role in the entire life cycle of a coronavirus [18,19,20]. Therefore, the N gene is an ideal target for designing shRNA to inhibit PDCoV replication. In this study, we constructed four plasmids expressing shRNA targeting the N gene and evaluated whether shRNA-mediated RNAi could inhibit PDCoV infection in vitro and in vivo. Surprisingly, stably expressed shRNAs almost completely blocked PDCoV reproduction in vitro, including shRNA-N1, N2, and N3; while the amount of viral genomic RNAs in the shRNA-N4 group decreased relatively less, which suggests a limited antiviral effect. For this phenomenon, one of the possible reasons is that the efficacy of shRNA-N4 was influenced by the secondary structure on target sites [17]. However, the regarding information has rarely been clarified. To enhance the antiviral effectiveness of the shRNA-N1 plasmid against PDCoV, a double-shRNA-expressing plasmid was further generated in this study. As assessed, pSil-double-shRNA-N1-mcherry plasmids significantly inhibited PDCoV replication both in vitro and in vivo, indicating that PDCoV infection could be suppressed by the combinatorial RNAi (co-RNAi) method [40].

Recombinant viral vector and shRNA-expressing plasmids are the currently common strategies for RNAi delivery into cells. The viral vector could be used to internalize specific shRNA into cells with high efficiency and obtain long-term inhibition. As non-viral vectors, shRNA-expressing plasmids cannot readily cross the membrane into cells and exist stably in vivo. Further studies are needed to mitigate this problem. Because of the rich synthesis of shRNA and the efficient RNAi induced by the human U6 (hU6) promoter, the pSil-double-shRNA-N1 plasmids bearing dual hU6 promoter/shRNA cassettes were established for the transcription of shRNA-N1. Nevertheless, excessive shRNA expression may cause cytotoxicity through competitively inhibiting endogenous miRNAs in cells [41]. In our work, there was no significant cytotoxicity in stably transfected LLC-PK1 cells. However, in the histopathological examination of the different sections of the small intestines of the piglets challenged with PDCoV, both the villus heights and crypt depths decreased in the piglets after shRNA administration alone, compared with the mock control. Lymphocyte proliferation could also be found in the lamina propria of the small intestines. These phenomena may indicate that shRNA-expressing plasmids yielded toxicity effects to some extent in the piglets. Hence, more attention should be paid to eliminate the adverse reaction of exogenous shRNA expression.

## 5. Conclusions

The present study evidenced that the double-shRNA-expressing plasmid synergistically targeting the N gene of PDCoV could effectively inhibit PDCoV replication and protect piglets from PDCoV-induced intestinal injury. Our preliminary results demonstrate that the co-RNAi platform might offer a candidate agent for PDCoV therapeutic applications, and also provide insight into the important role of the N gene in PDCoV replication.

## Figures and Tables

**Figure 1 animals-11-01216-f001:**
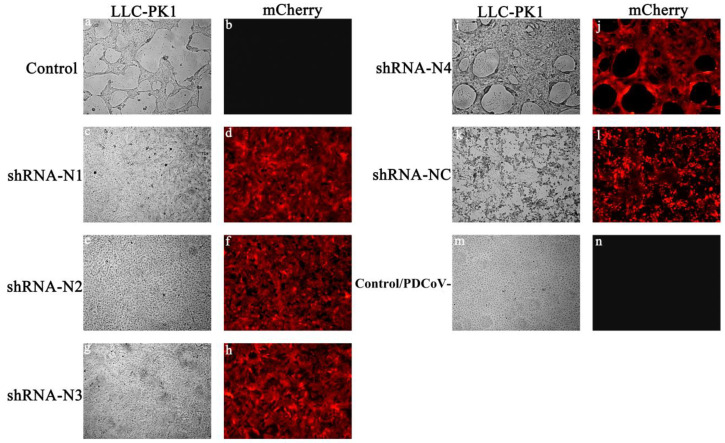
Fluorescence observation of CPE on LLC-PK1 cells transfected with shRNAs (100×). The LLC-PK1 cells were initially transfected with different plasmids carrying shRNA gene(s) and then infected with PDCoV at an MOI of 0.1, and CPE was observed at 36 hpi. (**a**,**b**) LLC-PK1 cells infected with PDCoV. (**c**–**l**) LLC-PK1 cells transfected with shRNA-N1, shRNA-N2, shRNA-N3, shRNA-N4, and shRNA-NC, respectively. (**m**,**n**) LLC-PK1 cells that were negative controls, that is, neither transfected with plasmids nor infected with virus.

**Figure 2 animals-11-01216-f002:**
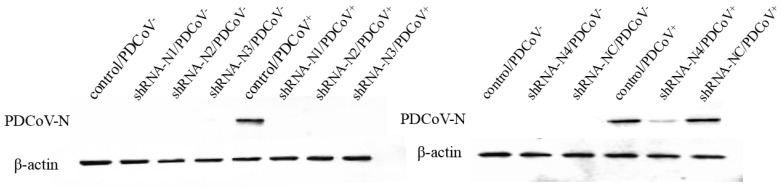
Expression of the N protein of PDCoV assessed by Western blotting. Equal amounts of cell lysates of PDCoV-infected/non-infected LLC-PK1 cells harvested at 36 hpi were tested using anti-PDCoV polyclonal antibodies (1:50 diluent), with β-actin as a protein reference control.

**Figure 3 animals-11-01216-f003:**
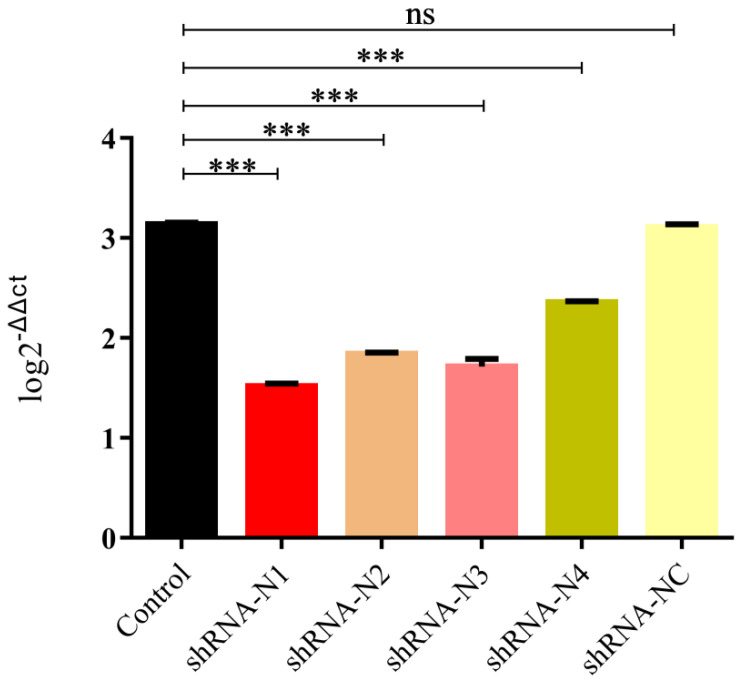
Inhibition of PDCoV replication by shRNAs in LLC-PK1 cells. qRT-PCR for the detection of the mRNA transcripts of the N gene of PDCoV relative to β-actin transcripts in the same sample. The data are presented as the mean values ± standard deviation for three independent experiments, and marked with three asterisks (***) if *p* < 0.001.

**Figure 4 animals-11-01216-f004:**
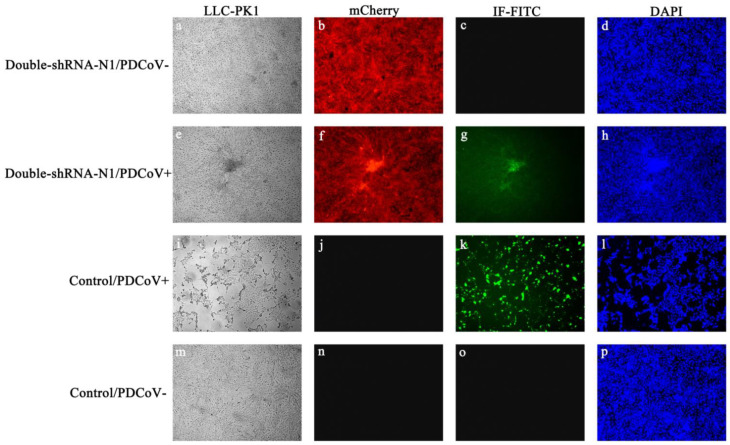
Effects of pSil-double-shRNA-N1-mcherry plasmids on PDCoV-induced CPE in LLC-PK1 cells by immunofluorescence assay. (**a**–**d**) pSil-double-shRNA-N1-mCherry stably transfected cells served as a mock transfection control; (**e**–**h**) pSil-double-shRNA-N1-mCherry stably transfected cells indicated a noticeable reduction of PDCoV N protein expression; (**i**–**l**) non-transfected cells indicated no obvious effects on PDCoV infection as a positive control; (**m**–**p**) non-transfected cells without PDCoV infection were used as a negative control. The green and red colors show the presence of PDCoV N protein-positive cells and stably shRNA-transfected cells, respectively. The nuclei of the cells were stained in blue by DAPI.

**Figure 5 animals-11-01216-f005:**
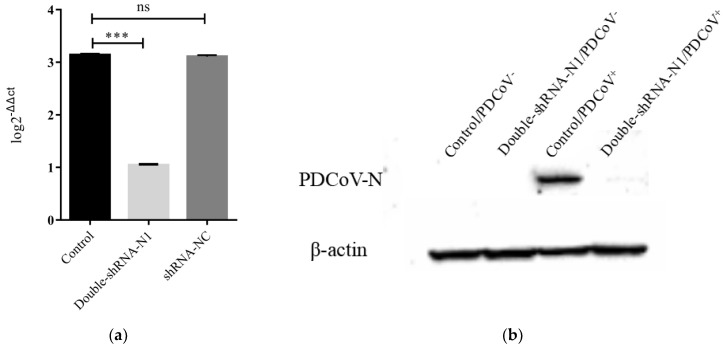
The reduction of PDCoV RNA and PDCoV N protein triggered by double-shRNA-N1-expressing plasmids. (**a**) qRT-PCR analysis of the inhibition of viral RNA synthesis in LLC-PK1 cells expressing double-shRNA-N1, the results are marked with three asterisks (***) if *p* < 0.001; (**b**) Western blot analysis of double-shRNA-N1-expressing plasmids inhibiting PDCoV N protein expression in LLC-PK1 cells at 36 hpi.

**Figure 6 animals-11-01216-f006:**
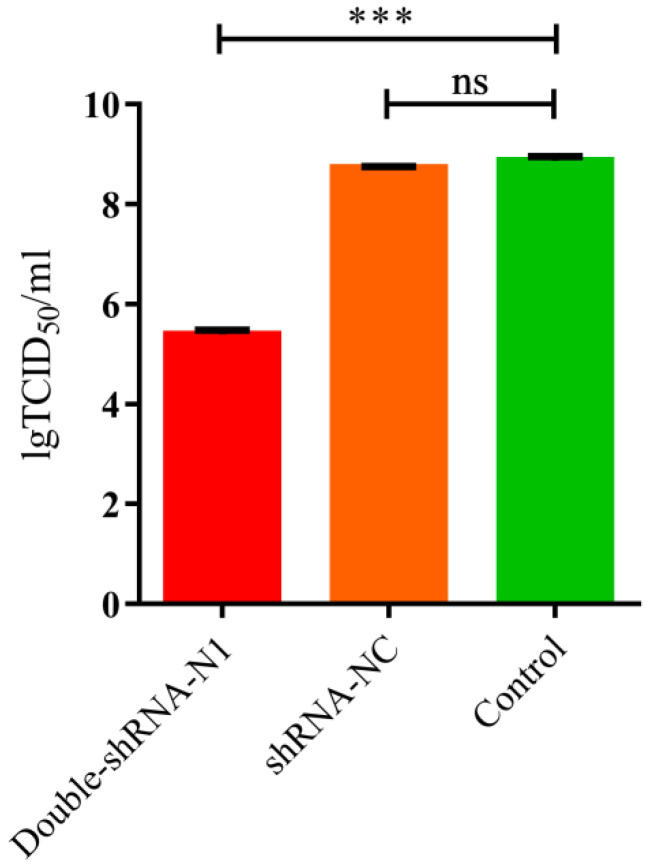
TCID_50_ determination of PDCoV on pSil-double-shRNA-N1-mCherry-transfected LLC-PK1 cells collected at 36 hpi. As indicated, the titer of PDCoV in pSil-double-shRNA-N1-mCherry-transfected LLC-PK1 cells is much lower than the other treatment groups of cells. Viral titers are shown by TCID_50_. The data are presented as the mean values ± standard deviation from three repeated experiments, and marked with three asterisks (***) if *p* < 0.001.

**Figure 7 animals-11-01216-f007:**
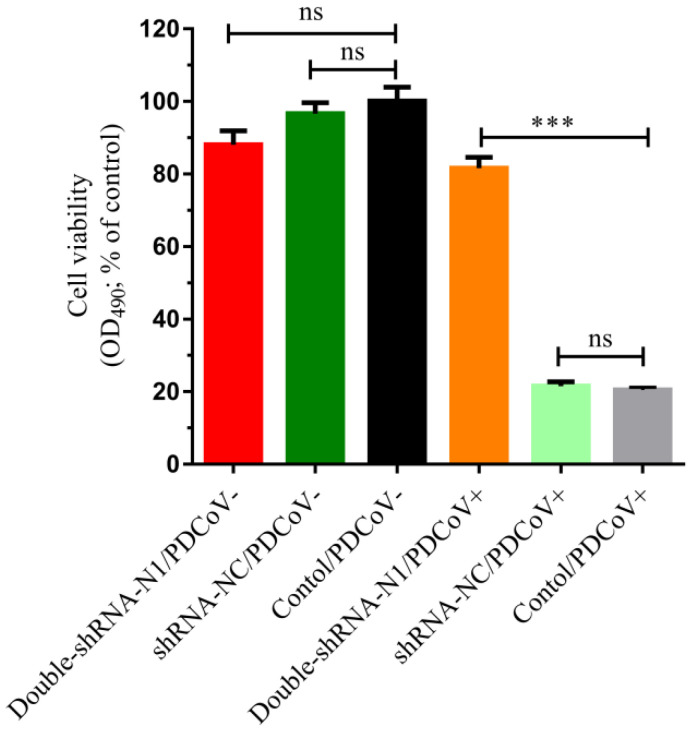
The viability observation of LLC-PK1 cells transfected with double-shRNA-N1-Mcherry during PDCoV infection. The cells were transfected with pSil-double-shRNA-N1-mCherry and then infected with PDCoV at an MOI of 0.1. At 36 hpi, the cell viability was calculated using the MTS assay, and the data are marked with three asterisks (***) if *p* < 0.001.

**Figure 8 animals-11-01216-f008:**
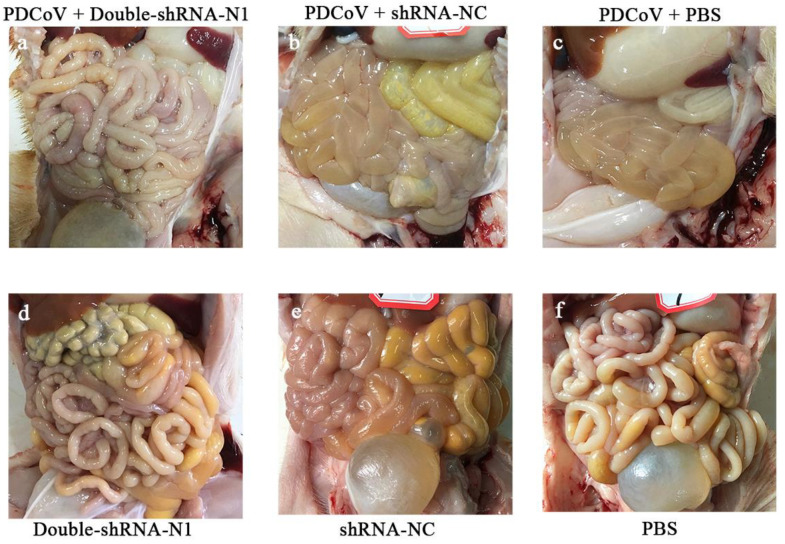
Effects of the double-shRNA-N1-expressing plasmids treatment on piglet intestinal changes. At 48 h post-challenge, all the piglets were euthanized and necropsied to examine the macroscopic lesions. No visible gross lesions in the small intestines were observed in the piglets in the P + V, P, N, and C groups (**a**,**d**–**f**). Meanwhile, the small intestinal walls of the piglets in the N + V and C + V groups were thin and transparent (**b**,**c**).

**Figure 9 animals-11-01216-f009:**
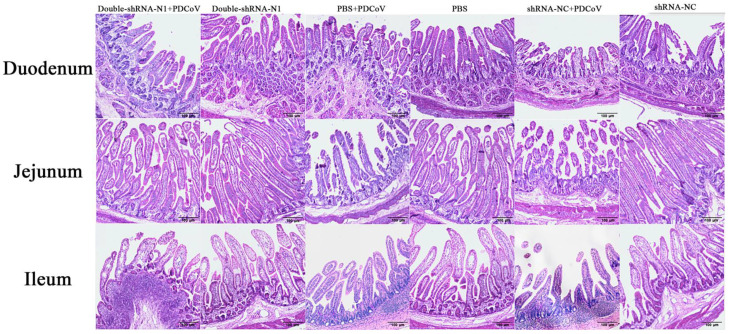
shRNA−expressing plasmids protect piglets from histopathological lesions induced by PDCoV. The small intestines of all piglets were collected for histopathological examination.

**Figure 10 animals-11-01216-f010:**
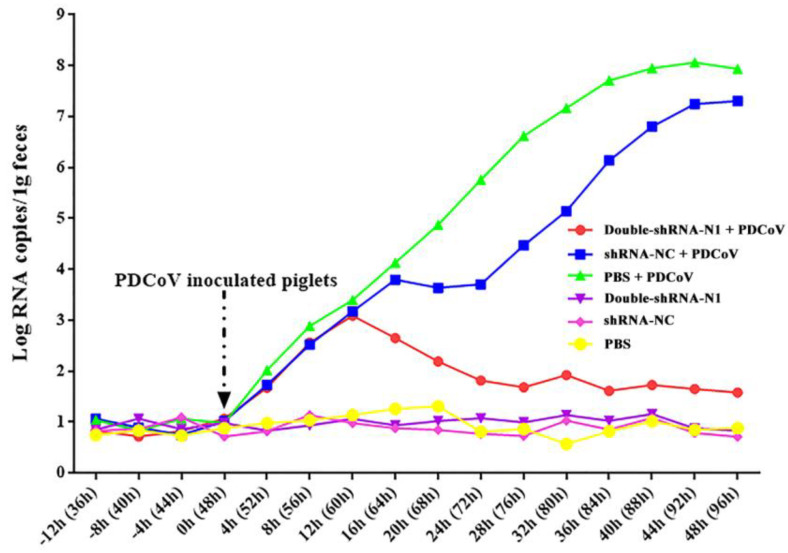
Effects of the double−shRNA−N1−expressing plasmids treatment in the PDCoV challenge model. The total RNAs from the fecal samples at different time points were extracted, and PDCoV replication kinetics in the piglets were determined by the qRT-PCR method. The PDCoV quantity was calculated and expressed as viral copies per 0.1 g feces, based on the standard curve.

**Table 1 animals-11-01216-t001:** The sequences and location of shRNAs targeting the N gene of PDCoV.

Name	Sequence (5′ to 3′)	Position
N1	5′-TTTAATAGAAGTGTCAGCC-3′	24305-24321
N2	5′-TAAATACCTGAGAAATGGC-3′	24594-24610
N3	5′-TGTTAACAGATTGAGATCC-3′	24468-24484
N4	5′-TATACTTAAGATTTCCTCC-3′	24225-24241
NC	5′-TAACATAGGGCAATTTAGC-3′	

**Table 2 animals-11-01216-t002:** Primers for quantitative real-time PCR.

Name	Sequence
N ^a^	F: 5′-ATCGACCACATGGCTCCAA-3′
	R: 5′-CAGCTCTTGCCCATGTAGCTT-3′
β-actin	F: 5′-CTCTTCCAGCCCTCCTTCC-3′
	R: 5′-GGTCCTTGCGGATGTCG-3′

Note: a: reference strain, CH/JXNI/02/2015 (GenBank accession number: KR131621.1).

**Table 3 animals-11-01216-t003:** Plasmids delivery and virus infection regimen.

	Group					
	P + V	P	N + V	N	C + V	C
Plasmid	pSil-double-shRNA-N1-mcherry	pSil-double-shRNA-N1-mcherry	pSil-shRNA-NC-mcherry	pSil-shRNA-NC-mcherry	PBS	PBS
Vector dosage	3 mg/1 mL/piglet	3 mg/1 mL/piglet	3 mg/1 mL/piglet	3 mg/1 mL/piglet	1 mL/piglet	1 mL/piglet
PDCoV	PDCoV	PBS	PDCoV	PBS	PDCoV	PBS
Viral dosage	10^5^ TCID_50_/10 mL/piglet	10 mL/piglet	10^5^ TCID_50_/10 mL/piglet	10 mL/piglet	10^5^ TCID_50_/10 mL/piglet	10 mL/piglet

**Table 4 animals-11-01216-t004:** Effects of the different treatments on villus height and crypt depth of the small intestines of the piglets.

	Group	V/C Ratio
Duodenum	P + V	6.56
	P	15.45
	N + V	2.76
	N	5.52
	C + V	1.12
	C	16.54
Jejunum	P + V	8.45
	P	11.38
	N + V	2.46
	N	11.79
	C + V	0.94
	C	13.97
Ileum	P + V	9.68
	P	11.46
	N + V	2.23
	N	12.86
	C + V	1.29
	C	13.82

Note: The V/C ratio was calculated from the data in the Appendix A.

## Data Availability

Not Applicable.

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
