# Peer review of "Plasmids Expressing shRNAs Specific to the Nucleocapsid Gene Inhibit the Replication of Porcine Deltacoronavirus In Vivo"

_animals, 2021, doi:10.3390/ani11051216_

Round 1

Reviewer 1 Report

Presented study is related to porcine deltacoronavirus (PDCoV) which may occure in pig herds, causing mortality of the animals as well as high economical losses in this kind of animal production.  The manuscript is important for the science and - what is more - for the livestock (pig) industry. It is also clearly written and I have no doubts that it should be published. My concerns are mostly related to the materials and methods, especially to the number of animals, what is the basis of this work. Why only 12 animals were used in presented report? Six groups with only two animals per each group makes the study not reliable and veracious. There is why I strongly suggest to clarify and describe why this number of animals in the text was taken. What is more, I would rather reccomend to change a title of the manuscript and transform it as a "preliminary" or "clinical" report. Moreover, the conlusions should be improved, being strictly related to the results of the study. As a summary, it was a pleasure to read this interesting manuscript, the Authors loadwork is significant in it (taking into account the text and the figures), but some minor revisions should be made to successfully publish this study in Animals. I wish you all the best! 

Author Response

We would like to thank you and reviewers for your critical reviews of our manuscript. The comments are all constructive and valuable for improving the quality of the manuscript. Based on the reviewers’ comments, we have made corresponding modifications by way of a point-by-point style.

Q: Presented study is related to porcine deltacoronavirus (PDCoV) which may occur in pig herds, causing mortality of the animals as well as high economical losses in this kind of animal production. The manuscript is important for the science and - what is more - for the livestock (pig) industry. It is also clearly written and I have no doubts that it should be published. My concerns are mostly related to the materials and methods, especially to the number of animals, what is the basis of this work. Why only 12 animals were used in presented report? Six groups with only two animals per each group makes the study not reliable and veracious. There is why I strongly suggest to clarify and describe why this number of animals in the text was taken. What is more, I would rather recommend to change a title of the manuscript and transform it as a "preliminary" or "clinical" report. Moreover, the conclusions should be improved, being strictly related to the results of the study. As a summary, it was a pleasure to read this interesting manuscript, the Authors loadwork is significant in it (taking into account the text and the figures), but some minor revisions should be made to successfully publish this study in Animals. I wish you all the best!

A: Thanks for reviewer’s valuable comments. The authors agree to the concerns raised by the reviewer, that is, the number of piglets used in this study was not enough for drawing a solid conclusion. We do wish that we would have used more piglets for animal challenge studies. However, unfortunately it was too difficult for us to obtain enough piglets for the experiments owing to the outbreaks of Africa swine fever (ASF) in China occurred since Aug, 2018, in which the experiments were performed. ASF, a devastating disease of swine, resulted in a sharp drop in pig populations, so that the strict limitations have been set by the authority of Chinese government on pig transportation in order to control the ASF epidemic. More terribly, in 2020, the outbreak of COVID-19 made it more difficult to repeat the experiment and validate our results. In addition, PDCoV-negative pigs are insufficient due to high prevalence of PDCoV in swine herds in China, which further adversely impact the re-implementation of our animal experiments. In the circumstances, we think it’s wise for us to quit the attempts for repetition with more piglets. Per reviewer’s suggestions, we have added “preliminary” before “results……” and “findings……” at page 1, line 67 and 80 and revised the conclusions of our manuscript. For details, please see the revised manuscript.

Reviewer 2 Report

This research describes a potential way of RNAi-based antivirals to control PDCoV infection. But the data display in some figures is not clear. Here are some comments:

Major concern:

  1. In figure 4, change “Control/PDCoV+” to “pSil-shRNA-NC-mCherry/ PDCoV+”. Also, explain why panel j has no mCherry expression.
  2. Line 256, what do the authors mean by saying “…an enhanced inhibition potential of double shRNA...” ? Do the authors want to compare it with single shRNA expressing plasmid? If yes, please include the data about single shRNA-N1 in figure 5, 6 and 7 for easy comparison. Please clarify.
  3. In figure 5, please include the data about shRNA-NC.
  4. Too many letters in Table 4. It’s better just to display the statistical analysis results about “V/C value” since the authors describe the results about “V/C ratio, not the villus height or crypt depth. The statistical data about villus height and crypt depth can be shown as the supplementary

Minor concern:

  1. For the non-transfected cells without PDCoV infection, the authors use “Normal” in figure 1, “Control” in figure 3-6, and “Mock” in figure 7, please choose one and use the same label in these figures.
  2. Line 163 and 164, change “qPCR” to “qRT-PCR” since the authors use “qRT-PCR” in line 252.
  3. Line 399, change “…reproduction” to “…reproduction in vitro”

Author Response

We would like to thank you and reviewers for your critical reviews of our manuscript. The comments are all constructive and valuable for improving the quality of the manuscript. Based on the reviewers’ comments, we have made corresponding modifications by way of a point-by-point style.

Major concern:

Q1. In figure 4, change “Control/PDCoV+” to “pSil-shRNA-NC-mCherry/ PDCoV+”. Also, explain why panel j has no mCherry expression.

A1. We are very sorry for the mistake in Figure 4. The non-transfected cells were wrongly labelled as pSil-shRNA-NC-mCherry stably transfected cells in figure annotation due to our ignorance. We have made corresponding corrections.

Q2. Line 256, what do the authors mean by saying “…an enhanced inhibition potential of double shRNA...” ? Do the authors want to compare it with single shRNA expressing plasmid? If yes, please include the data about single shRNA-N1 in figure 5, 6 and 7 for easy comparison. Please clarify. In figure 5, please include the data about shRNA-NC.

A2. Thanks for the constructive suggestions. Yes, compared with the qRT-PCR results of the N gene copy number as a surrogate of PDCoV replication as shown in Figure 3 and 5 in the manuscript, double shRNA-N1 plasmid showed more potent inhibitory activity on PDCoV RNA synthesis than single shRNA-N1 plasmid, and we added this portion in the supplementary Figure S1. According to reviewer’s suggestions, we have made corresponding modifications, including adding shRNA-NC in Fig 5.

Q3. Too many letters in Table 4. It’s better just to display the statistical analysis results about “V/C value” since the authors describe the results about “V/C ratio, not the villus height or crypt depth. The statistical data about villus height and crypt depth can be shown as the supplementary

A3. Response: Thank you for review’s useful suggestions, we have made corresponding modifications in the Table 4 and the statistical data about villus height and crypt depth are provided as the supplementary material. For details, please refer to the revised manuscript.

Minor concern:

Q1. For the non-transfected cells without PDCoV infection, the authors use “Normal” in figure 1, “Control” in figure 3-6, and “Mock” in figure 7, please choose one and use the same label in these figures.

Line 163 and 164, change “qPCR” to “qRT-PCR” since the authors use “qRT-PCR” in line 252.

Line 399, change “…reproduction” to “…reproduction in vitro”

A1. Thanks so much for the constructive suggestions. We have made corresponding modifications per reviewer’s comments. For details, please refer to the revised manuscript with tracking changes.

Reviewer 3 Report

This manuscript explored the potential of silencing nucleocapsid protein expression as a means to inhibit PDCoV replication and pathogenesis. The major finding of the study was that one of the four shRNAs tested can significantly inhibit viral replication in cell culture and in piglets (with two tandem copies of the shRNA). The experimental design and conclusion are generally well supported by the data presented. Having said that, I would suggest the authors to perform experiments to directly compare the inhibitory effects of shRNA#1 when delivered in single copy or double copy at least in cell culture. From the data presented, it is not immediately clear that double copy is significantly more effective than single copy (Figures 3, 5, and 6). This is very relevant as the authors rightly discussed that excessive shRNA may be associated with cytotoxicity (lines 416-417). 

Two minor points:

  1. When discussing RNAi delivery by viral vectors, it should be specified which viral vector that can confer long-term inhibition.
  2. The manuscript needs editing by a native speaker because some of the sentences sound a bit strange.  

Author Response

We would like to thank you for your critical reviews of our manuscript. The comments are all constructive and valuable for improving the quality of the manuscript. Based on the reviewers’ comments, we have made corresponding modifications by way of a point-by-point style.

A1. This manuscript explored the potential of silencing nucleocapsid protein expression as a means to inhibit PDCoV replication and pathogenesis. The major finding of the study was that one of the four shRNAs tested can significantly inhibit viral replication in cell culture and in piglets (with two tandem copies of the shRNA). The experimental design and conclusion are generally well supported by the data presented. Having said that, I would suggest the authors to perform experiments to directly compare the inhibitory effects of shRNA#1 when delivered in single copy or double copy at least in cell culture. From the data presented, it is not immediately clear that double copy is significantly more effective than single copy (Figures 3, 5, and 6). This is very relevant as the authors rightly discussed that excessive shRNA may be associated with cytotoxicity (lines 416-417). 

A1. Thanks for the constructive suggestions. Yes, compared with the qRT-PCR results of the N gene copy number as a surrogate of PDCoV replication as shown in Figure 3 and 5 in the manuscript, double shRNA-N1 plasmid showed more potent inhibitory activity on PDCoV RNA synthesis than single shRNA-N1 plasmid, and we added this portion in the supplementary Figure S1.

Two minor points:

Q1. When discussing RNAi delivery by viral vectors, it should be specified which viral vector that can confer long-term inhibition.

A1. This is a great suggestion. We have added the corresponding modifications in the discussion.

Q2. The manuscript needs editing by a native speaker because some of the sentences sound a bit strange. 

A2. Thanks for your advice. Per your request, the manuscript has been revised and proofread by a person with full professional proficiency in English.

Round 2

Reviewer 2 Report

This revised version has improved quality of presentation. Just one minor comment here:  Line 264, change ‘than” to “as compared with the samples from”

Author Response

we sincerely want to thank you and reviewers for your critical reviews and great inputs.

Q: This revised version has improved quality of presentation. Just one minor comment here:  Line 264, change ‘than” to “as compared with the samples from”

A: Thanks for your good suggestion, we have made the corresponding correction in the revised manuscript.

Reviewer 3 Report

Authors have addressed most of my comments. However, no change has been made to properly discuss RNAi delivery by viral vectors - my minor point #1, although the authors stated that modifications were made in the discussion. 

Author Response

We sincerely want to thank you for your critical reviews and great inputs

Authors have addressed most of my comments. However, no change has been made to properly discuss RNAi delivery by viral vectors - my minor point #1, although the authors stated that modifications were made in the discussion.

Thanks for constructive suggestion. I am sorry that maybe our tracks in the last revised manuscript make you confused. We have added “Especially, viral vectors including lentiviruses, adenovirus, and other self-replicating RNA viruses could trigger long-term inhibition activity [34]” in Line 373-374 and a reference “34. Lundstrom, K. Viral Vectors Applied for RNAi-Based Antiviral Therapy. Viruses 2020, 12, 924.” In Line 518 according to the PDF file designated “animals-1162932-peer-review-r1”. We hope the revised manuscript would meet the criteria.